# Improving Question Generation with Multi-level Content Planning

**Zehua Xia[1], Qi Gou[1], Bowen Yu[2], Haiyang Yu[2], Fei Huang[2]**
**Yongbin Li[2]\* and Cam-Tu Nguyen[1]\***
[1]State Key Laboratory for Novel Software Technology, Nanjing University, China
[2]Alibaba Group
{zehuaxia, qi.gou}@smail.nju.edu.cn
{yubowen.ybw, yifei.yhy, f.huang}@alibaba-inc.com
shuide.lyb@alibaba-inc.com ncamtu@nju.edu.cn

## Abstract

This paper addresses the problem of generating questions from a given context and an answer, specifically focusing on questions that require multi-hop reasoning across an extended context. Previous studies have suggested that key phrase selection is essential for question generation (QG), yet it is still challenging to connect such disjointed phrases into meaningful questions, particularly for long context. To mitigate this issue, we propose MultiFactor, a novel QG framework based on multi-level content planning. Specifically, MultiFactor includes two components: *FA-model*, which simultaneously selects key phrases and generates full answers, and *Q-model* which takes the generated full answer as an additional input to generate questions. Here, full answer generation is introduced to connect the short answer with the selected key phrases, thus forming an answer-aware summary to facilitate QG. Both *FA-model* and *Q-model* are formalized as simple-yet-effective Phrase-Enhanced Transformers, our joint model for phrase selection and text generation. Experimental results show that our method outperforms strong baselines on two popular QG datasets. Our code is available at https://github.com/zeaver/MultiFactor.

## 1 Introduction

Question Generation (QG) is a crucial task in the field of Natural Language Processing (NLP) which focuses on creating human-like questions based on a given source context and a specific answer. In recent years, QG has gained considerable attention from both academic and industrial communities due to its potential applications in question answering (Duan et al., 2017), machine reading comprehension (Du et al., 2017), and automatic conversation (Pan et al., 2019; Ling et al., 2020).

Effective content planning is essential for QG sys-

---
\*Corresponding authors.

---

| |
|---|
| **Source Paragraph 1: Richard Hornsby & Sons** |
| Richard Hornsby & Sons was an engine and machinery manufacturer in Lincolnshire, England from 1828 until 1918. The company was a pioneer in the manufacture of the oil engine developed by Herbert Akroyd Stuart, which was marketed under the "Hornsby-Akroyd" name. The company developed an early track system for vehicles, selling the patent to Holt & Co. (predecessor to Caterpillar Inc.) in America. In 1918, Richard Hornsby & Sons became a subsidiary of the neighbouring engineering firm Rustons of Lincoln, to create "Ruston & Hornsby". |
| **Source Paragraph 2: Herbert Akroyd Stuart** |
| Herbert Akroyd-Stuart (28 January 1864, Halifax, Yorkshire, England – 19 February 1927, Halifax) was an English inventor who is noted for his invention of the hot bulb engine, or heavy oil engine. Akroyd-Stuart was born in Halifax, Yorkshire, but lived in Australia for a period in his early years. He was educated at Newbury Grammar School (now St. Bartholomew's School) and Finsbury Technical College in London. |
| **Gold Question:** What is the date of birth of the English inventor that developed the Richard Hornsby & Sons oil engine? |
| **Answer:** 28 January 1864 |
| **Vanilla QG:** When was the English inventor Herbert Akroyd Stuart born? |
| **Phrase-level QG:** When was the English inventor who developed the oil engine born? |
| **MultiFactor:** When was the English inventor who developed the oil engine pioneered by Richard Hornsby & Sons born? |
| **Generated Full Answer:** The English inventor who developed the oil engine pioneered by Richard Hornsby & Sons born in 28 January 1864. |

Figure 1: An example from HotpotQA in which the question generated by MultiFactor requires reasoning over disjointed facts across documents.

tems to enhance the quality of the output questions. This task is particularly important for generating complex questions, that require reasoning over long context. Based on the content granularity, prior research (Zhang et al., 2021) can be broadly categorized into two groups: phrase-level and sentence-level content planning. On one hand, the majority of prior work (Sun et al., 2018; Liu et al., 2019; Pan et al., 2020; Cao and Wang, 2021; Fei et al., 2022; Subramanian et al., 2018) has focused on phrase-level planning, where the system identifies key phrases in the context and generates questions based on them. For instance, given the answer "28 January 1864" and a two-paragraphs context in Figure 1, we can recognize "English Inventor," "the oil engine," "Herbert Akroyd Stuart" as important text for generating questions. For long context, however, it is still challenging for machines to connect such disjointed facts to form meaningful questions. On the other hand, sentence-

level content planning, as demonstrated by Du and Cardie (2017), aims at automatic sentence selection to reduce the context length. For instance, given the sample in Figure 1, one can choose the underscored sentences to facilitate QG. Unfortunately, it is observable that the selected sentences still contain redundant information that may negatively impact question generation. Therefore, we believe that an effective automatic content planning at both the phrase and the sentence levels is crucial for generating questions.

In this paper, we investigate a novel framework, MultiFactor, based on multi-level content planning for QG. At the fine-grained level, answer-aware phrases are selected as the focus for downstream QG. At the coarse-grained level, a full answer generation is trained to connect such (disjointed) phrases and form a complete sentence. Intuitively, a full answer can be regarded as an answer-aware summary of the context, from which complex questions are more conveniently generated. As shown in Figure 1, MultiFactor is able to connect the short answer with the selected phrases, and thus create a question that requires more hops of reasoning compared to Vanilla QG. It is also notable that we follow a generative approach instead of a selection approach (Du and Cardie, 2017) to sentence-level content planning. Figure 1 demonstrates that our generated full answer contains more focused information than the selected (underscored) sentences.

Specifically, MultiFactor includes two components: 1) A *FA-model* that simultaneously selects key phrases and generate full answers; and 2) A *Q-model* that takes the generated full answer as an additional input for QG. To realize these components, we propose Phrase-Enhanced Transformer (PET), where the phrase selection is regarded as a joint task with the generation task both in *FA-model* and *Q-model*. Here, the phrase selection model and the generation model share the Transformer encoder, enabling better representation learning for both tasks. The selected phrase probabilities are then used to bias to the Transformer Decoder to focus more on the answer-aware phrases. In general, PET is simple yet effective as we can leverage the power of pretrained language models for both the phrase selection and the generation tasks.

Our main contributions are summarized as follows:

- To our knowledge, we are the first to introduce

the concept of full answers in an attempt of multi-level content planning for QG. As such, our study helps shed light on the influence of the answer-aware summary on QG.

- We design our MultiFactor framework following a simple yet effective pipeline of Phrase-enhanced Transformers (PET), which jointly model the phrase selection task and the text generation task. Leveraging the power of pretrained language models, PET achieves high effectiveness while keeping the additional number of parameters fairly low in comparison to the base model.

- Experimental results validate the effectiveness of MultiFactor on two settings of HotpotQA, a popular benchmark on multi-hop QG, and SQuAD 1.1, a dataset with shorter context.

## 2 Related Work

Early Question Generation (QG) systems (Mostow and Chen, 2009; Chali and Hasan, 2012; Heilman, 2011) followed a rule-based approach. This approach, however, suffers from a number of issues, such as poor generalization and high-maintenance costs. With the introduction of large QA datasets such as SQuAD (Rajpurkar et al., 2016) and Hot-potQA (Yang et al., 2018), the neural-based approach has become the mainstream in recent years. In general, these methods formalize QG as a sequence-to-sequence problem (Du et al., 2017), on which a number of innovations have been made from the following perspectives.

**Enhanced Input Representation** Recent question generation (QG) systems have used auxiliary information to improve the representation of the input sequence. For example, Du et al. (2017) used paragraph embeddings to enhance the input sentence embedding. Du and Cardie (2018) further improved input sentence encoding by incorporating co-reference chain information within preceding sentences. Other studies (Su et al., 2020; Pan et al., 2020; Fei et al., 2021; Sachan et al., 2020a) enhanced input encoding by incorporating semantic relationships, which are obtained by extracting a semantic or entity graph from the corresponding passage, and then applying graph attention networks (GATs) (Veličković et al., 2018).

One of the challenges in QG is that the model might generate answer-irrelevant questions, such as pro-

ducing inappropriate question words for a given answer. To overcome this issue, different strategies have been proposed to effectively exploit answer information for input representation. For example, Zhou et al. (2017); Zhao et al. (2018); Liu et al. (2019) marked the answer location in the input passage. Meanwhile, Song et al. (2018); Chen et al. (2020) exploited complex passage-answer interaction strategies. Kim et al. (2019); Sun et al. (2018), on the other hand, sought to avoid answer-included questions by using separating encoders for answers and passages. Compared to these works, we also aim to make better use of answer information but we do so from the new perspective of full answers.

**Content Planning** The purpose of content planning is to identify essential information from context. Content planning is widely used in in text generation tasks such as QA/QG, dialogue system (Fu et al., 2022; Zhang et al., 2023; Gou et al., 2023), and summarization (Chen et al., 2022). Previous studies (Sun et al., 2018; Liu et al., 2019) predicted "clue" words based on their proximity to the answer. This approach works well for simple QG from short contexts. For more complex questions that require reasoning from multiple sentences, researchers selected entire sentences from the input (documents, paragraphs) as the focus for QG, as in the study conducted by Du and Cardie (2017). Nevertheless, coarse-grained content planning at the sentence level may include irrelevant information. Therefore, recent studies (Pan et al., 2020; Fei et al., 2021, 2022) have focused on obtaining finer-grained information at the phrase level for question generation. In these studies, semantic graphs are first constructed through dependency parsing or information extraction tools. Then, a node classification module is leveraged to choose essential nodes (phrases) for question generation.

Our study focuses on content planning for Question Generation (QG) but differs from previous studies in several ways. Firstly, we target automatic content-planning at both the fine-grained level of phrases and the coarse-grained level of sentences. As far as we know, we are the first that consider multiple levels of granularity for automatic content planning. Secondly, we propose a novel phrase-enhanced transformer (PET) which is a simple yet effective for phrase-level content planning. Compared to Graph-based methods, PET is relatively simpler as it eliminates the need for semantic graph

construction. In addition, PET is able to leverage the power of pre-trained language models for its effectiveness. Thirdly, we perform content planning at the sentence level by following the generative approach instead of the extraction approach as presented in the study by Du and Cardie (2017). The example in Figure 1 shows that our generated full answer contains less redundant information than selecting entire sentences of supported facts.

**Diversity** While the majority of previous studies focus on generating context-relevant questions, recent studies (Cho et al., 2019; Wang et al., 2020b; Fan et al., 2018; Narayan et al., 2022) have sought to improve diversity of QG. Although we not yet consider the diversity issue, our framework provides a convenient way to improve diversity while maintaining consistency. For example, one can perform diverse phrase selection or look for diverse ways to turn full answers into questions. At the same time, different strategies can be used to make sure that the full answer is faithful to the given context, thus improving the consistency.

## 3 Methodology

### 3.1 MultiFactor Question Generation

Given a source context $c = [w_1, w_2, \dots, w_{T_c}]$ and an answer $a = [a_1, a_2, \dots, a_{T_a}]$, the objective is to generate a relevant question $q = [q_1, q_2, \dots, q_{T_q}]$; where $T_c$, $T_a$, and $T_q$ denote the number of tokens in $c$, $a$ and $q$, respectively. It is presumed that we can generate full answers $s = [s_1, s_2, \dots, a_{T_s}]$ of $T_s$ tokens, thus obtaining answer-relevant summaries of the context. The full answers are subsequently used for generating questions as follows:

$$p(q|c, a) = \mathbb{E}_s[\underbrace{p(q|s, c, a)}_{\text{Q model}} \underbrace{p(s|c, a)}_{\text{FA model}}] \quad (1)$$

where *Q model* and *FA model* refer to the question generation and the full answer generation models, respectively. Each *Q-model* and *FA-model* is formalized as a Phrase-enhanced Transformer (PET), our proposal for text generation with phrase planning. In the following, we denote a PET as $\phi : x \to y$, where $x$ is the input sequence and $y$ is the output sequence. For the *FA-model*, the input sequence is $x = c \oplus a$ and the output is the full answer $s$, where $\oplus$ indicates string concatenation. As for the *Q-model*, the input is $x = c \oplus a \oplus s$ with $s$ being the best full answer from *FA-model*, and the output is the question $q$. The PET model $\phi$

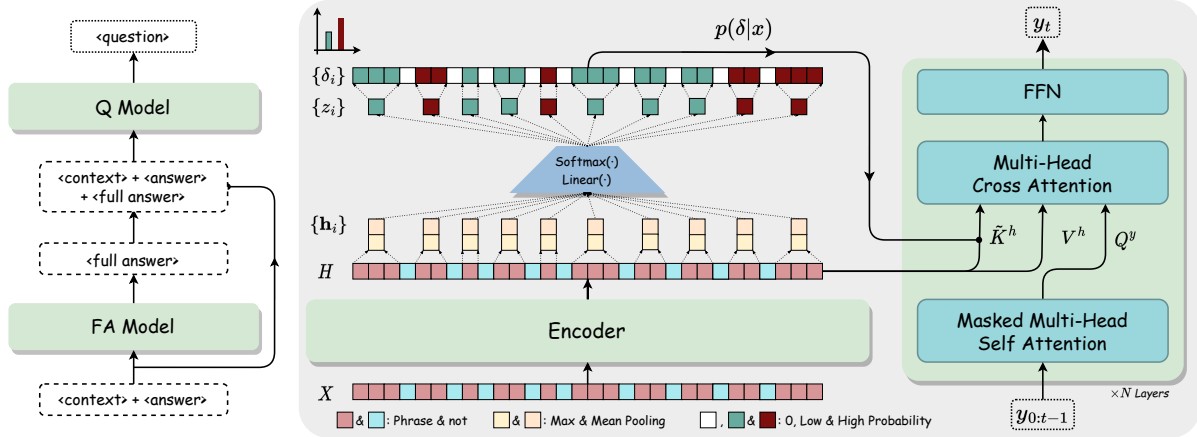

Figure 2: Overview of our MultiFactor is shown on the left. Here, *FA-model* and *Q-model* share the same architecture of Phrase-Enhanced Transformer demonstrated on the right.

firsts select phrases that can consistently be used to generate the output, then integrates the phrase probabilities as soft constraints for the decoder to do generation. The overview of MultiFactor is demonstrated in Figure 2. The Phrase-Enhanced Transformer is detailed in the following section.

### 3.2 Phrase Enhanced Transformer

We propose Phrase-enhanced Transformer (PET), a simple yet effective Transformer-based model to infuse phrase selection probability from encoder into decoder to improve question generation.

Formally, given the input sequence $x$, and $L$ phrase candidates, the $i$-th phrase $w_i^z$ ($i \in \{1, ..., L\}$) is a sequence of $L_i$ tokens $\{w_{l_1^i}^z, w_{l_2^i}^z, \ldots, w_{l_{L_i}^i}^z\}$ extracted from the context $x$, where $l_j^i$ indicates the index of token $j$ of the $i$-th phrase in $x$. The phrase-level content planning is formalized as assigning a label $z_i \in [0, 1]$ to each phrase in the candidate pool, where $z_i$ is 1 if the phrase should be selected and 0 otherwise. The phrase information is then integrated to generate $y$ auto-regressively:

$$p(y|x, z) = \prod_{t=1}^{T_y} p(y_t|x, z, y_{0:t-1})$$

**Encoder and Phrase Selection**    Recall that the input $x$ contains the context $c$ and the answer $a$ in both *Q-model* and *FA-model*, and thus we select the candidate phrases only from the context $c$ by extracting entities, verbs and noun phrases using SpaCy[1]. The phrase selection is formalized as a binary classification task, where the input is a phrase

[1]https://spacy.io/

encoding obtained from the transformer encoder:

$$H = \text{Encoder}(x)$$
$$\mathbf{h}_i^z = \text{MeanMaxPooling}(\{H_j\}_{j=l_1^i}^{l_{L_i}^i})$$
$$z_i = \text{Softmax}\{\text{Linear}[\mathbf{h}_i^z]\}$$

where $H \in \mathcal{R}^{T_x \times d}$ with $T_x$ and $d$ being the length of input sequence and dimensions of hidden states, respectively. Here, Encoder indicates the Transformer encoder, of which the details can be found in (Devlin et al., 2019). The phrase representation $\mathbf{h}_i^z$ is obtained by concatenating $\text{MaxPooling}(\cdot)$ and $\text{MeanPooling}(\cdot)$ of the hidden states $\{H_j\}$ corresponding to i-th phrase. We then employ a linear network with $\text{Softmax}(\cdot)$ as the phrase selection probability estimator (Galke and Scherp, 2022).

**Probabilistic Fusion in Decoder**    Decoder consumes previously generated tokens $y_{1...t-1}$ then generates the next one as follows:

$$y_t = \text{Softmax}[\text{Linear}[\text{DecLayers}(y_{1...t-1}, H)]]$$

where $H$ is the Encoder output, and DecLayers indicates a stack of $N$ decoder layers. Like Transformer, each PET decoder layer contains three sublayers: 1) the masked multi-head attention layer; 2) the multi-head cross-attention layer; 3) the fully connected feed-forward network. Considering the multi-head cross-attention sublayer is the interaction module between the encoder and decoder, we modify it to take into account the phrase selection probability $z_i$ as shown in Figure 2.

Here, we detail the underlying mechanism of each cross-attention head and how we modify it to encode phrase information. Let us recall that the

input for a cross-attention layer includes a query state, a key state, and a value state. The query state $Q^y$ is the (linear) projection of the output of the first sublayer (the masked multi-head attention layer). Intuitively, $Q^y$ encapsulates the information about the previously generated tokens. The key state $K^h = HW^k$ and the value state $V^h = HW^v$ are two linear projections of the Encoder output $H$. $W^k \in \mathcal{R}^{d \times d_k}$ and $W^v \in \mathcal{R}^{d \times d_v}$ are the layer parameters, where $d_k$ and $d_v$ are the dimensions of the key and value states. The output of the cross-attention layer is then calculated as follows:

$$\text{CrossAtten}(Q, K, V) = \text{Softmax}(\frac{QK^\top}{\sqrt{d_k}})V$$

Here, we drop the superscripts for simplicity, but the notations should be clear from the context. Theoretically, one can inject the phrase information to either $V^h$ or $K^h$. In practice, however, updating the value state introduces noises that counter the effect of pretraining Transformer-based models, which are commonly used for generation. As a result, we integrate phrase probabilities to the key state, thus replacing $K^h$ by a new key state $\tilde{K}^h$:

$$\tilde{K}^h = \delta W^\delta + HW^k$$

$$\delta_j = \begin{cases} [1 - z_i, z_i], & \text{if } j \text{ in } i-\text{th phrase} \\ [1, 0], & j \text{ not in any phrase} \end{cases}$$

where $W^\delta \in \mathcal{R}^{2 \times d_k}$ is the probabilistic fusion layer. Here, $z_i$ is the groundtruth phrase label for phrase $i$ during training ($z_i \in \{0, 1\}$), and the predicted probabilities to select the i-th phrase during inference ($z_i \in [0, 1]$). In *Q-model*, we choose all tokens $w_i$ in the full answer $s$ as important tokens.

**Training**  Given the training data set of triples $(x, z, y)$, where $x$ is the input, $y$ is the groundtruth output sequence and $z$ indicates the labels for phrases that can be found in $y$, we can simultaneously train the phrase selection and the text generation model by optimizing the following loss:

$$\mathcal{L} = \text{CrossEntropy}[\hat{y}, y] + \lambda \, \text{CrossEntropy}[\hat{z}, z]$$

where $\hat{z}$ is the predicted labels for phrase selection, $\hat{y}$ is the predicted output, $\lambda$ is a hyper-parameter.

## 4  Experiments

### 4.1  Experimental Setup

**Datasets**  We evaluate our method on two different QG tasks: a complex task on HotpotQA and

| Dataset | HotpotQA | | SQuAD 1.1 |
| | Sup. | Full | |
|---|---|---|---|
| Context Len. | 49.3 | 210.7 | 26.8 |
| Question Len. | 18.0 | 18.0 | 10.9 |
| Train/Dev/Test | 89947/500/7405 | | 86635/8965/8964 |

Table 1: The statistics of HotpotQA and SQuAD 1.1, where Supp. and Full indicate the supporting facts setting and the full setting of HotpotQA.

a simpler task on SQuAD 1.1. There are two settings for HotpotQA (see Table 1): 1) HotpotQA (sup. facts) where the sentences that contain supporting facts for answers are known in advance; 2) HotpotQA (full) where the context is longer and contains several paragraphs from different documents. For SQuAD 1.1, we use the split proposed in Zhou et al. (2017). Although our MultiFactor is expected to work best on HotpotQA (full), we consider HotpotQA (sup. facts) and SQuAD 1.1 to investigate the benefits of multi-level content planning for short contexts.

**Metrics**  Following previous studies, we exploit commonly-used metrics for evaluation, including BLEU (Papineni et al., 2002), METEOR (Banerjee and Lavie, 2005) and ROUGE-L (Lin, 2004). We also report the recently proposed BERTScore (Zhang et al., 2020) in the ablation study.

**Implementation Details**  We exploit two base models for MultiFactor: T5-base [2] and MixQG-base [3]. To train *FA-model*, we apply QA2D (Demszky et al., 2018) to convert question and answer pairs to obtain pseudo (gold) full answers. Both *Q-model* and *FA-model* are trained with $\lambda$ of 1. Our code is implemented on Huggingface (Wolf et al., 2020), whereas AdamW (Loshchilov and Hutter, 2019) is used for optimization. More training details and data format are provided in Appendix B.

**Baselines**  The baselines (in Table 2) can be grouped into several categories: 1) Early seq2seq methods that use GRU/LSTM and attention for the input representation, such as SemQG, NGQ++, and s2sa-at-mcp-gsa; 2) Graph-based methods for content planning like ADDQG, DP-Graph, IGND, CQG, MulQG, GATENLL+CT, Graph2seq+RL; 3) Pretrained-language models based methods, including T5-base, CQG, MixQG, and QA4QG. Among these baselines, MixQG and QA4QG are strong

---

[2]https://huggingface.co/t5-base
[3]https://huggingface.co/Salesforce/mixqg-base

| | Model | B-1 | B-2 | B-3 | B-4 | MTR | R-L |
|---|---|---|---|---|---|---|---|
| | *Encoder Input: Supporting Facts Sentences* | | | | | | |
| **HotpotQA** | SemQG (Zhang and Bansal, 2019) | 39.92 | 26.73 | 18.73 | 14.71 | 19.29 | 35.63 |
| | ADDQG (Wang et al., 2020a) | 44.34 | 31.32 | 22.68 | 17.54 | 20.56 | 38.09 |
| | F+R+A (Xie et al., 2020) | 37.97 | - | - | 15.41 | 19.61 | 35.12 |
| | DP-Graph (Pan et al., 2020) | 40.55 | 27.21 | 20.13 | 15.53 | 20.15 | 36.94 |
| | IGND (Fei et al., 2021) | 41.22 | 24.71 | 18.99 | 16.36 | 24.19 | 38.34 |
| | T5-base (Raffel et al., 2020) | 47.78 | 36.39 | 29.44 | 24.48 | 25.59 | 43.17 |
| | CQG (Fei et al., 2022) | 49.71 | 37.04 | 29.93 | 25.09 | 27.45 | 41.83 |
| | MixQG-base (Murakhovs'ka et al., 2022)† | 49.60 | 37.78 | 30.58 | 25.45 | 26.36 | 43.21 |
| | QA4QG-large (Su et al., 2022) | 49.55 | 37.91 | 30.79 | 25.70 | 27.44 | **46.48** |
| | MultiFactor (T5-base) | 53.46 | 40.95 | 33.29 | 27.80 | 28.26 | 43.80 |
| | MultiFactor (MixQG-base) | **54.17** | **41.50** | **33.74** | **28.22** | **28.60** | 44.17 |
| | *Encoder Input: Full Document Context* | | | | | | |
| | MulQG (Su et al., 2020) | 40.15 | 26.71 | 19.73 | 15.20 | 20.51 | 35.30 |
| | GATENLL+CT (Sachan et al., 2020b) | - | - | - | 20.02 | 22.40 | 39.49 |
| | T5-base (Raffel et al., 2020) | 42.68 | 31.67 | 25.21 | 20.70 | 22.57 | 40.25 |
| | MixQG-base (Murakhovs'ka et al., 2022) † | 45.28 | 33.72 | 26.90 | 22.13 | 23.78 | 41.21 |
| | QA4QG-large (Su et al., 2022) | 46.45 | 33.83 | 26.35 | 21.21 | 25.53 | 42.44 |
| | MultiFactor (T5-base) | 51.41 | 39.31 | 31.90 | 26.66 | 29.66 | 43.37 |
| | MultiFactor (MixQG-base) | **54.84** | **42.41** | **34.69** | **29.12** | **30.01** | **45.20** |
| **SQuAD 1.1** | NQG++ (Zhou et al., 2017) | 42.46 | 26.33 | 18.46 | 13.51 | - | - |
| | s2sa-at-mcp-gsa (Zhao et al., 2018) | 44.51 | 29.07 | 21.06 | 15.82 | 19.67 | 44.24 |
| | APM (Sun et al., 2018) | 43.02 | 28.14 | 20.51 | 15.64 | - | - |
| | Graph2seq+RL (Chen et al., 2020) | - | - | - | 18.30 | 21.70 | 45.98 |
| | T5-base (Raffel et al., 2020) | 47.96 | 33.58 | 25.54 | 20.15 | 24.21 | 40.33 |
| | IGND (Fei et al., 2021) | **50.82** | 34.73 | 25.64 | 20.33 | - | **48.94** |
| | MixQG-base (Murakhovs'ka et al., 2022)† | 49.69 | 35.19 | 26.70 | 21.44 | 25.48 | 41.22 |
| | MultiFactor (T5-base) | 49.56 | 35.00 | 26.78 | 21.24 | **25.63** | 41.22 |
| | MultiFactor (MixQG-base) | 50.51 | **35.78** | **27.42** | **21.75** | 25.55 | 41.62 |

Table 2: Automatic evaluation results on HotpotQA (Yang et al., 2018) and SQuAD 1.1 (Rajpurkar et al., 2016). The **Bold** and underline mark the best and second-best results. The B-x, MTR, and R-L mean BLEU-x, METEOR, and ROUGE-L, respectively. We mark the results reproduced by ourselves with † , other results are from Fei et al. (2022), Su et al. (2022) and Fei et al. (2021).

ones with QA4QG being the state-of-the-art model on HotpotQA. Here, MixQG is a pretrained model tailored for the QG task whereas QA4QG exploits a Question Answering (QA) model to enhance QG.

## 4.2 Main Results

The performance MultiFactor and baselines are shown in Table 2 with the following main insights.

On HotpotQA, it is observable that our method obtains superior results on nearly all evaluation metrics. Specifically, MultiFactor outperforms the current state-of-the-art model QA4QG by about 8 and 2.5 BLEU-4 points in the full and the supporting facts setting, respectively. Note that we achieve such results with a smaller number of model parameters compared to QA4QG-large. Specifically, the current state-of-the-art model exploits two BART-

large models (for QA and QG) with a total number of parameters of 800M, whereas MultiFactor has a total number of parameters of around 440M corresponding to two T5/MixQG-base models. Here, the extra parameters associated with phrase selection in PET (T5/MixQG-base) is only 0.02M, which is relatively small compared to the number of parameters in T5/MixQG-base.

By cross-referencing the performance of common baselines (MixQG or QA4QG) on HotpotQA (full) and HotpotQA (supp. facts), it is evident that these baselines are more effective on HotpotQA (supp. facts). This is intuitive since the provided supporting sentences can be regarded as sentence-level content planning that benefits those on HotpotQA (supp. facts). However, even without this advantage, MultiFactor on HotpotQA (full.) outperforms

| Model | B-4 | MTR | R-L | BSc |
|---|---|---|---|---|
| HotpotQA ( Supporting Facts) | | | | |
| Fine-tuned | 25.45 | 26.36 | 43.21 | 51.49 |
| Cls+Gen | 25.90 | 26.73 | 43.55 | 52.04 |
| One-hot PET-Q | 27.48 | 28.28 | 43.46 | 52.63 |
| PET-Q | 27.79 | 28.46 | 43.94 | 53.05 |
| MultiFactor | **28.22** | **28.60** | **44.17** | **53.44** |
| HotpotQA (Full Document) | | | | |
| Fine-tuned | 22.13 | 23.78 | 41.21 | 48.76 |
| Cls+Gen | 22.39 | 23.95 | 41.40 | 48.95 |
| One-hot PET-Q | 26.61 | 28.94 | 43.11 | 52.21 |
| PET-Q | 26.82 | 29.04 | 43.53 | 52.58 |
| MultiFactor | **29.12** | **30.01** | **45.20** | **54.49** |
| SQuAD 1.1 | | | | |
| Fine-tuned | 19.96 | 24.39 | 39.77 | 55.31 |
| Cls+Gen | 20.14 | 24.45 | 39.83 | 55.34 |
| One-hot PET-Q | 21.10 | 25.35 | 41.10 | 56.52 |
| PET-Q | 21.33 | 25.38 | 41.58 | 56.89 |
| MultiFactor | **21.75** | **25.55** | **41.62** | **56.93** |

Table 3: The ablation study for MultiFactor (**MixQG-base**), MultiFactor (T5-base) is shown in Appendix C.

these baselines on HotpotQA (supp. facts), showing the advantages of MultiFactor for long context.

On SQuAD, MultiFactor is better than most baselines on multiple evaluation metrics, demonstrating the benefits of multi-level content planning even for short-contexts. However, the margin of improvement is not as significant as that seen on HotpotQA. MultiFactor falls behind some baselines, such as IGND, in terms of ROUGE-L. This could be due to the fact that generating questions on SQuAD requires information mainly from a single sentence. Therefore, a simple copy mechanism like that used in IGND may lead to higher ROUGE-L.

### 4.3 Ablation Study

We study the impact of different components in MultiFactor and show the results with MixQG-base in Table 3 and more details in Appendix C. Here, "Fine-tuned" indicates the MixQG-base model, which is finetuned for our QG tasks. For **Cls+Gen**, the phrase selection task and the generation task share the encoder and jointly trained like in PET. The phrase information, however, is not integrated into the decoder for generation, just to enhance the encoder. **One-hot PET-Q** indicates that instead of using the soft labels (probabilities of a phrase to be selected), we use the predicted hard labels (0 or 1) to inject into PET. And finally,

| Model | B-4 | MTR | R-L | BSc |
|---|---|---|---|---|
| PET-Q | 27.45 | 28.28 | 43.46 | 52.41 |
| MultiFactor | 27.80 | 28.26 | 43.80 | 52.86 |
| *Q-model* | | | | |
| *w/o* Context | 27.63 | 28.13 | 43.66 | 52.69 |
| *w* Oracle-FA | 31.61 | 29.66 | 48.84 | 56.15 |
| *w* Gold-FA | 91.08 | 64.10 | 93.00 | 93.77 |

Table 4: Results on MultiFactor (T5-base) and its variants on HotpotQA (supp. facts).

**PET-Q** denotes MultiFactor without the full answer information.

**Phrase-level Content Planning**   By comparing PET-Q, one-hot PET-Q and Cls+Gen to the fine-tuned MixQG-base in Table 3, we can draw several observations. First, adding the phrase selection task helps improve QG performance. Second, integrating phrase selection to the decoder (in One-hot PET-Q and PET-Q) is more effective than just exploiting phrase classification as an additional task (as in Cls+Gen). Finally, it is recommended to utilize soft labels (as in PET-Q) instead of hard labels (as in One-hot PET-Q) to bias the decoder.

**Sentence-level Content Planning**   By comparing MultiFactor to other variants in Table 3, it becomes apparent that using the full answer prediction helps improve the performance of QG in most cases. The contribution of the FA-model is particularly evident in HotpotQA (full), where the context is longer. In this instance, the FA-model provides an answer-aware summary of the context, which benefits downstream QG. In contrast, for SQuAD where the context is shorter, the FA-model still helps but its impact appears to be less notable.

### 4.4 The Roles of Q-model and FA-model

We investigate two possible causes that may impact the effectiveness of MultiFactor, including potential errors in converting full answers to questions in *Q-model*, and error propagation from the *FA-model* to the *Q-model*. For the first cause, we evaluate *Q-model* (w/ Gold-FA), which takes as input the gold full answers, rather than *FA-model* outputs. For the second cause, we assess *Q-model* (w/o Context) and *Q-model* (w/ Oracle-FA). Here, *Q-model* (w/ Oracle-FA) is provided with the oracle answer, which is the output with the highest BLEU among the top five outputs of *FA-model*.

Table 4 reveals several observations on HotpotQA

(supp. facts) with MultiFactor (T5-base). Firstly, the high effectiveness of *Q-model* (with Gold-FA) indicates that the difficulty of QG largely lies in the full answer generation. Nevertheless, we can still improve *Q-model* further, by, e.g., predicting the question type based on the grammatical role of the short answer in *FA-model* outputs. Secondly, *Q-model* (w/o Context) outperforms PET-Q but not MultiFactor. This might be because context provides useful information to mitigate the error propagation from *FA-model*. Finally, the superior of *Q-model* (with Oracle-FA) over MultiFactor shows that the greedy output of *FA-model* is suboptimal, and thus being able to evaluate the top *FA-model* outputs can help improve overall effectiveness.

## 4.5 Human Evaluation

Automatic evaluation with respect to one gold question cannot account for multiple valid variations that can be generated from the same input context/answer. As a result, three people were recruited to evaluate four models (T5-base, PET-Q, MultiFactor and its variant with Oracle-FA) on 200 random test samples from HotpotQA (supp. facts). Note that the evaluators independently judged whether each generated question is correct or erroneous. In addition, they were not aware of the identity of the models in advance. In the case of an error, evaluators are requested to choose between two types of errors: hop errors and semantic errors. Hop errors refer to questions that miss key information needed to reason the answer, while semantic errors indicate questions that disclose answers or is nonsensical. Additionally, we analyse the ratio of errors in two types of questions on HotpotQA: *bridge*, which requires multiple hops of information across documents, and *comparison*, which often starts with "which one" or the answer is of yes/no type. Human evaluation results are shown in Table 5, and we also present some examples in Appendix E.

**MultiFactor vs Others** Comparing MultiFactor to other models (T5, PET-Q) in Table 5, we observe an increase in the number of correct questions, showing that multi-level content planning is effective. The improvement of MultiFactor over PET-Q is more noticeable in contrast with that in Table 4 with automatic metrics. This partially validates the role of full answers even with short contexts. In such instances, full answers can be seen as an answer-aware paraphrase of the context that is more convenient for downstream QG. In addition,

| Model | T5 | PET-Q | Multi | Ocl-FA |
|---|---|---|---|---|
| Correct | 83.5 | 86.0 | 87.5 | 89.5 |
| Hop Error | 11.5 | 9.5 | 9.0 | 7.5 |
| Semantic Error | 5.0 | 4.5 | 3.5 | 3.0 |
| Error (Bridge) | 13.5 | 11.0 | 9.0 | 7.0 |
| Error (Comparison) | 3.0 | 3.0 | 3.5 | 3.5 |

Table 5: Human evaluation results on HotpotQA (supp. facts), where Multi and Ocl-FA indicates MultiFactor (T5-base) and its variant where *Q-model* is given the oracle full answer (w/ Oracle-FA). The last two lines show the error rates where questions are of bridge or comparison types.

one can see a significant reduction of semantic error in MultiFactor compared to PET-Q. This is because the model better understands how a short answer is positioned in a full answer context, as such we can reduce the disclosure of (short) answers or the wrong choice of question types. However, there is still room for improvement as MultiFactor (w/ Oracle-FA) is still much better than the one with the greedy full answer from *FA-model* (referred to as Multi in Table 5). Particularly, there should be a significant reduction in hop error if one can choose better outputs from *FA-model*.

**Error Analysis on Question Types** It is observable that multi-level content planning plays important roles in reducing errors associated with "bridge" type questions, which is intuitive given the nature of this type. However, we do not observe any significant improvement with comparison type. Further examination reveals two possible reasons: 1) the number of this type of questions is comparably limit; 2) QA2D performs poorly in reconstructing the full answers for this type. Further studies are expected to mitigate these issues.

## 4.6 Comparison with LLM-based QG

As Large Language Model (LLM) performs outstandingly in various text generation tasks, we evaluate the performance of GPT-3.5 zero-shot[4] (Brown et al., 2020) and LoRA fine-tuned Llama2-7B (Hu et al., 2022; Touvron et al., 2023) on HotpotQA (full document). Implementation details regarding instructions and LoRA hyper-parameters are provided in Appendix B.

**Automatic Evaluation** The performance of Llama2-7B and GPT-3.5-Turbo (zero-shot) in comparison with MultiFactor, T5-base (finetuned) and

---

[4]Via the Azure OpenAI Service.

| Model | B-4 | MTR | R-L | BSc |
|---|---|---|---|---|
| MultiFactor | | | | |
| *w.* T5-base | 26.66 | 29.66 | 43.37 | 52.76 |
| *w.* MixQG-base | 29.12 | 30.01 | 45.20 | 54.49 |
| T5-base | 20.70 | 22.57 | 40.25 | 44.06 |
| MixQG-base | 22.13 | 23.78 | 41.21 | 48.76 |
| Llama2-7B-LoRA | 16.53 | 21.35 | 33.03 | 37.44 |
| GPT-3.5-Turbo | | | | |
| *w. zero*-shot | 8.78 | 14.84 | 22.48 | 28.38 |

Table 6: The automatic scores of GPT-3.5 zero-shot, LoRA fine-tuned Llama2-7B on HotpotQA full document setting.

| Model | Win | Tie | Lose |
|---|---|---|---|
| Llama2-7B-LoRA | | | |
| *v.s.* T5-base | 20 | 60 | 20 |
| *v.s.* MultiFactor (T5-base) | 13 | 65 | 22 |
| GPT-3.5-Turbo *w. zero*-shot | | | |
| *v.s.* MultiFactor (T5-base) | 20 | 29 | 51 |

Table 7: Human evaluation on GPT-3.5 zero-shot and LoRA fine-tuned Llama2-7B in comparison with Multi-Factor (T5-base).

MixQG-base (finetuned) are given in Table 6, where several observations can be made. Firstly, MultiFactor outperforms other methods on automatic scores by a large margin. Secondly, finetuning results in better automatic scores comparing to zero-shot in-context learning with GPT-3.5-Turbo. Finally, Llama2-7B-LoRA is inferior to methods that are based on finetuning moderate models (T5-base/MixQG-base) across all of these metrics.

**Human Evaluation** As LLM tend to use a wider variety of words, automatic scores based on one gold question do not precisely reflect the quality of these models. As a result, we conducted human evaluation and showed the results on Table 7. Since OpenAI service may regard some prompts as invalid (i.e. non-safe for work), the evaluation was conducted on 100 valid samples from the sample pool that we considered in Section 4.5. The human annotators were asked to compare a pair of methods on two dimensions, the factual consistency and complexity. The first dimension is to ensure that the generated questions are correct, and the second dimension is to prioritize complicated questions as it is the objective of multi-hop QG.

Human evaluation results from Table 7 show that human annotators prefer MultiFactor (T5-base) to Llama2-7B-LoRA and GPT-3.5-Turbo (zero-shot). Additionally, Llama2-7b-LoRA outperforms GPT-3.5-Turbo (zero-shot), which is consistent with the automatic evaluation results in Table 6. Interestingly, although T5-base (finetuning) outperforms Llama2-7B-LoRA in Table 6, human evaluation shows that these two methods perform comparably. The low automatic scores for Llama2-7B-LoRA are due to its tendency to rephrase outputs instead of copying the original context. Last but not least,

in-depth analysis also reveals a common issue with GPT-3.5-Turbo (zero-shot): its output questions often reveal the given answers. Therefore, multi-level content planning in instruction or demonstration for GPT-3.5-Turbo could be used to address this issue in LLM-based QG, potentially resulting in better performance.

## 5   Conclusion and Future Works

This paper presents MultiFactor, a novel QG method with multi-level content planning. Specifically, MultiFactor consists of a *FA-model*, which simultaneously select important phrases and generate an answer-aware summary (a full answer), and *Q-model*, which takes the generated full answer into account for question generation. Both *FA-model* and *Q-model* are formalized as our simple yet effective PET. Experiments on HotpotQA and SQuAD 1.1 demonstrate the effectiveness of our method.

Our in-depth analysis shows that there is a lot of room for improvement following this line of work. On one hand, we can improve the full answer generation model. On the other hand, we can enhance the *Q-model* in MultiFactor either by exploiting multiple generated full answers or reducing the error propagation.

## 6   Limitations

Our work may have some limitations. First, the experiments are only on English corpus. The effectiveness of MultiFactor is not verified on the datasets of other languages. Second, the context length in sentence-level QG task is not very long as shown in Table 8. For particularly long contexts ($> 500$ or 1000), it needs more explorations.

## 7   Ethics Statement

MultiFactor aims to improve the performance of the answer-aware QG task, especially the complex QG. During our research, we did not collect any other datasets, instead conduct our experiments and construct the corresponding full answer on these previously works. Our generation is completely within the scope of the datasets. Even the result is incorrect, it is still controllable and harmless, no potential risk. The model is currently English language only, whose practical applications is limited in the real world.

## 8   Acknowledgements

We would like to thank the anonymous reviewers for their insightful comments. We also like to thank Dr. Jingyang Li for their helpful suggestions. This work was supported by Alibaba Innovative Research project "Document Grounded Dialogue System".

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

## A  Statistic of Datasets

Here, we list the length of context, question and answer of the HotpotQA and SQuAD 1.1 datasets in Table 8. HotpotQA supporting facts and full document settings share the same output and semi-gold full answers.

| | Train | Valid | Test |
|---|---|---|---|
| HotpotQA (Supporting Facts Sentence) | | | |
| Context | 221/12/49.31 | 107/16/48.67 | 170/13/50.24 |
| Question | 89/4/18.07 | 80/6/17.58 | 43/7/16.30 |
| Answer | 69/1/2.35 | 15/1/2.39 | 30/1/2.62 |
| Phrase | 1.86/8.66 | 1.73/8.61 | 1.57/9.14 |
| FA | 82579/89947 | 459/500 | 6763/7405 |
| HotpotQA (Full Document) | | | |
| Context | 2331/29/210.72 | 690/41/205.15 | 1371/35/216.68 |
| Question | 89/4/18.07 | 80/6/17.58 | 43/7/16.30 |
| Answer | 69/1/2.35 | 15/1/2.39 | 30/1/2.62 |
| Phrase | 7.46/36.49 | 7.35/35.43 | 6.63/37.27 |
| FA | 82579/89947 | 459/500 | 6763/7405 |
| SQuAD 1.1 | | | |
| Context | 285/2/26.83 | 150/2/27.61 | 150/4/27.47 |
| Question | 38/1/10.94 | 31/1/11.01 | 28/3/11.05 |
| Answer | 34/1/3.24 | 27/1/3.43 | 30/1/3.44 |
| Phrase | 2.34/6.80 | 3.75/7.06 | 3.75/7.05 |
| FA | 84976/86635 | 8864/8965 | 8840/8964 |

Table 8: The statistic of max/min/mean token length from NLTK tokenizer, the number of positive/negative phrases and the number of valid/total full answer(FA) examples in HotpotQA and SQuAD 1.1 datasets.

## B  Implementation Details

**Model Details**  MixQG pre-trained series models are fine-tuned from T5, having the same architecture and number of parameters. In addition to basic modules, MultiFactor adds a classifier ($2d \times 2$) and $L_d$ probability infusion layers ($2 \times d$), where $d$, $L_d$ donate the model dimensions and the number of decoder layers. Specifically, when initializing with T5-base (220M, $d = 768, L_d = 12$), MultiFactor only increases the number of parameters by $1536 \times 2 + 12 \times 2 \times 768 \approx 0.02$M (~0.01%).

**Training Details**  Because we train the model with fixed epochs on HotpotQA and the dev size is too small (500), we select the best result on test dataset directly following the previous work (Pan et al., 2020; Su et al., 2022) on HotpotQA. On SQuAD 1.1, we select the result based on the dev set. Max length of HotpotQA-full is 512, two others is 256. Moreover, the learning rate for MixQG-base is lower than that of the normal T5-base, as stated in (Murakhovs'ka et al., 2022). As a result, we have opted to employ learning rates of 5e-5 and 2e-5 for MixQG-base on HotpotQA and SQuAD 1.1, respectively, while T5-base are 1e-4 and 5e-5. All the batchsize is 32, except that HotpotQA-full is 16, where the training epoch is 5 instead of 10. We turn off the sampling, and beam size are 1 and 5 on HotpotQA and SQuAD 1.1, respectively. Others parameters are default value in Huggingface trainer and generator configuration files. More parameters and time cost of training and inference are in Table 9.

**Data Format**  We list the input formats of these experiments mentioned before in Table 10. And we use special tokens: <ans>, <passage>, <fa> to present the answer, context, and full answer start tokens.

**Instructions and LoRA hyper-parameters**  The instruction of zero-shot/Flan-T5-base/Llama2-7B is shown in Figure 3. As for LoRA fine-tuned hyper-parameters, we follow the llama-recipes[5] default settings, where $r = 28, \alpha = 32$.

## C  Ablation Study on T5

Considering T5 is a more general Text2Text Pretrained Lanuague Model, we also conduct ablation

---

[5]https://github.com/facebookresearch/llama-recipes

```
zero_shot_instrcution = f"Given the context and
    answer, please help me generate a multi-hop
    question.\\nAnswer: {{answer}}\\nContext: {{
    context}}\\nQuestion:"
```

Figure 3: The zero-shot instruction shown in python code.

| Dataset | HotpotQA | | SQuAD 1.1 |
| | Sup. | Full | |
|---|---|---|---|
| Learning rate | 5e-5/1e-4 | 5e-5 /1e-4 | 2e-5/5e-5 |
| Batch size | 16 | 32 | 32 |
| Warm-up ratio | 0.1 | 0.1 | 0.1 |
| Epochs | 5 | 10 | 15 |
| Training time | 20 | 20 | 18 |
| Inference time | 16 | 15 | 15 |
| Beam size | 1 | 1 | 5 |

Table 9: Details of training and inference. Data in learning rate is (MixQG-base/T5-base). The unit of training and inference time is `min/(epoch·GPU_num)`.

studies on T5-base, and the results are shown in Table 11.

# D  Ablation Study on Flan-T5

We conducted experiments initialized with Flan-T5-base to evaluate the performance of instruaction-finetuning model on HotpotQA full document setting. Results are shown in Table 12. Instruction is shown in Figure 3. Corss compared with these results in Table 3 and 11, Flan-T5-base outperforms T5-base significantly but still worse than MixQG-base. MixQG is a QG-specific pre-trained model and fine-tuned on nine various answer-type QA datasets from the T5-base. These results are line with our expectations.

# E  Error Examples

We list some error examples shown in Figure 4. In hop error, we show three types of hop errors: wrong hop, missing hop, and fabricating information, respectively. In semantic error, we list a declarative generation instead of a question and a nonsensical case in which the output is longer than the input. Lastly, we present a comparison type where both the pseudo gold and generated full answer are wrong, although almost comparison-type QA has no pseudo gold full answer.

| Type | Input |
|---|---|
| *FA-model* | <ans> {answer} <passage> {context} |
| *Q-model* | |
| T5 | <ans> {answer} <fa> {fa} <passage> {context} |
| *w/o* Context | <ans> {answer} <fa> {full_answer} |
| MixQG | {answer} /n <fa> {fa} <passage> {context} |
| PET | |
| T5 | <ans> {answer} <passage> {context} |
| MixQG | {answer} /n <passage> {context} |

Table 10: Input formats in our experiments.

| Model | B-4 | MTR | R-L | BSc |
|---|---|---|---|---|
| HotpotQA (Supporting Facts) | | | | |
| Fine-tuned | 24.48 | 25.59 | 43.17 | 50.93 |
| Cls+Gen | 25.36 | 26.33 | 43.38 | 51.49 |
| One-hot PET-Q | 27.01 | 28.11 | 42.91 | 52.31 |
| PET-Q | 27.45 | **28.28** | 43.46 | 52.41 |
| MultiFactor | **27.80** | 28.26 | **43.80** | **52.86** |
| HotpotQA (Full Document) | | | | |
| Fine-tuned | 20.70 | 22.57 | 40.25 | 44.06 |
| Cls+Gen | 20.81 | 22.61 | 40.58 | 44.24 |
| One-hot PET-Q | 25.94 | 28.75 | 43.10 | 51.63 |
| PET-Q | 26.35 | 29.54 | 43.08 | 52.33 |
| MultiFactor | **26.66** | **29.66** | **43.37** | **52.76** |
| SQuAD 1.1 | | | | |
| Fine-tuned | 20.15 | 24.21 | 40.33 | 55.18 |
| Cls+Gen | 20.29 | 24.27 | 40.34 | 55.22 |
| One-hot PET-Q | 20.31 | 25.49 | 40.43 | 56.06 |
| PET-Q | 21.13 | 25.34 | 41.03 | 56.21 |
| MultiFactor | **21.24** | **25.63** | **41.22** | **56.55** |

Table 11: The ablation study for MultiFactor, where the B-4, MTR, R-L and BSc means BLEU-4, METEOR, ROUGE-L and BERTScore, respectively.

| Model | B-4 | MTR | R-L | BSc |
|---|---|---|---|---|
| Fine-tuned | 21.69 | 23.31 | 40.82 | 47.68 |
| MultiFactor | 28.82 | 29.14 | 44.87 | 53.67 |

Table 12: The ablation study on Flan-T5-base on HotpotQA full document setting.

**Error Examples**

**Facts**:

i. 2015 Accra floods. Mayor of Accra Metropolitan Assembly, Alfred Oko Vanderpuije described the flooding as critical.

ii. 2015 Accra floods. At least 25 people have died from the flooding directly, while a petrol station explosion caused by the flooding killed at least 200 more people.

iii. 2015 Accra explosion. On June 4, 2015, an explosion and a fire occurred at a petrol station in Ghana's capital city Accra, killing over 250 people.

**Answer**: an explosion and a fire occurred at a petrol station

**Gold Question**: what caused the death of over 250 people in Accra, Ghana?

**Generated Question**: What happened at a petrol station in Ghana's capital city Accra on June 4, 2015, that killed over 250 people and caused a flood in the Accra Metropolitan Assembly, Alfred Oko Vanderpuije described the flooding as critical?

**Error Analysis**: hop error. A wrong hop, the explosion and a fire did not cause a flood.

**Facts**:

i. Jacksonville station. It serves the "Silver Meteor" and "Silver Star" trains as well as the Thruway Motorcoach to Lakeland.

ii. Silver Star (Amtrak train). The Silver Star is a 1522 mi passenger train route in the "Silver Service" brand operated by Amtrak, running from New York City south to Miami, Florida via the Northeast Corridor to Washington, D.C., then via Richmond, Virginia; Raleigh, North Carolina; Columbia, South Carolina; Savannah, Georgia; Jacksonville, Florida; Orlando, Florida; and Tampa, Florida.

**Answer**: 1522

**Gold Question**: How many miles does the train, which passes through the Amtrak Jacksonville station and shares the track with the Silver Meteor, run?

**Generated Question**: Jacksonville station serves the train that is how many miles long?

**Error Analysis**: Hop error. Missing a key information or key hop. The Jacksonville station servers at least 2 trains: "Silver Meteor" and "Silver Star". The generated question does not directly ask the "Silver Star", which runs 1522 miles.

**Facts**:

i. West Oaks, Houston. West Oaks is a small subdivision in Houston, Texas

ii. George H. W. Bush. George Herbert Walker Bush (born June 12, 1924) is an American politician who was the 41st President of the United States from 1989 to 1993 and the 43rd Vice President of the United States from 1981 to 1989.

**Answer**: 41st

**Gold Question**: George H. W. Bush, a resident of West Oaks Subdivision in Houston Texas, became what number President in 1989?

**Generated Question**: What number president of the United States was the man who owns West Oaks, Houston?

**Error Analysis**: Hop error. Fabricating the information that Bush owned West Oaks, Houston. Note, this error is caused by the insufficient input information. The source text does not indicate that Bush was a resident of West Oaks Subdivision in Houston Texas.

**Facts**:

i. Value premium. Other experts, such as John C. Bogle, have argued that no value premium exists, claiming that Fama and French's research is period dependent.

ii. John C. Bogle. He is the founder and retired chief executive of The Vanguard Group.

**Answer**: The Vanguard Group

**Gold Question**: In regards to Value premium an expert argued that no value premium exists, what did this expert found?

**Generated Question**: Value premium is a term used to describe the value of a product over other terms, other experts, such as John C. Bogle, have argued that no value premium exists, claiming that Fama and French's research is period dependent, Bogle is an American business executive, philanthropist, and former CEO, Bogle is the founder and retired chief executive of which organization?

**Error Analysis**: Semantic error. The question is even longer than the input, which is considered as a nonsensical generation. Because it utilizes the knowledge stored in parameters and ignore the input source information, which possibly leads hallucinatio.

**Facts**:

i. The Story of Them Featuring Van Morrison. The Story of Them Featuring Van Morrison is a compilation album that includes almost every song recorded by the Northern Irish band Them during the two-year history of the band when it featured Van Morrison as the vocalist for the group.

ii. Them (band). Them were a Northern Irish band formed in Belfast in April 1964, most prominently known for the garage rock standard \"Gloria\" and launching singer Van Morrison's musical career.

**Answer**: Them

**Gold Question**: The Story of Them Featuring Van Morrison is a compilation album recorded by what Northern Irish band formed in Belfast in April 1964?

**Generated Question**: The Story of Them Featuring Van Morrison is a compilation album that includes almost every song recorded by them Northern Irish band, most prominently known for the garage rock standard "Gloria" and launching singer Van Morrison's musical career?

**Error Analysis**: Semantic error. In fact, the generation is a declarative sentence. And the The Story of Them Featuring Van Morrison did not launch singer Van Morrison's musical career, which is also an error hop.

**Facts**:

i. In These Times. In These Times is an American politically progressive/democratic socialist monthly magazine of news and opinion published in Chicago, Illinois.

ii. Multinational Monitor. The Multinational Monitor was a bimonthly magazine founded by Ralph Nader in 1980.

**Answer**: Multinational Monitor

**Gold Question**: Which magazine has more issues each month, In These Times or Multinational Monitor?

**Pseudo-gold Full Answer:** Multinational Monitor has more issues each month, In These Times or Multinational Monitor.

**Generated Question**: Which magazine was founded first, In These Times or Multinational Monitor?

**Generated Full Answer:** Multinational Monitor was founded first, In These Times or Multinational Monitor.

**Error Analysis**: Hop error. Both input facts indicate the magazine publication frequency attribute but no established time. Only a few comparison-type QA pair was constructed a full answer successfully. This example is an exception, providing wrong sentence-level planning and generating a lousy question.

Figure 4: We show six representative error examples, which includes three hop error, two semantic error and one typical comparison-type error cases.