# OpenReview forum: "Improving Question Generation with Multi-level Content Planning"
_EMNLP/2023/Conference — EMNLP 2023 Findings_

### Official Review · Reviewer_SaFE · 2023-08-05

**Soundness:** 2

**Excitement:**

3: Ambivalent: It has merits (e.g., it reports state-of-the-art results, the idea is nice), but there are key weaknesses (e.g., it describes incremental work), and it can significantly benefit from another round of revision. However, I won't object to accepting it if my co-reviewers champion it.

**Paper Topic And Main Contributions:**

This paper studies the problem of generating questions from a given context and answer. To this end, the paper proposes MultiFactor, including a FA-model and a Q-model. Here, full answer generation is introduced to connect the short answer with the selected key phrases. Experimental results show that the method outperforms strong baselines on two popular dataset. Overall, this paper is easy to follow.

**Questions For The Authors:**

Please refer to "paper weakness", particularly W1 and W3.

**Reasons To Accept:**

S1. This paper is the first study that introduces the concept of full answers in an attempt at multi-level content planning for question generation (QG).
S2. The approach is well explained and authors well guide the reader through both the methods as well as experiments. Overall, this paper is easy to follow. Both the introduction and organization are great. Besides, the title, abstract, and keywords are consistent with the content of the paper.

**Reasons To Reject:**

W1. Authors state that "we do not evaluate the performance of LLM (e.g., ChatGPT or GPT-4) due to the limitation of accessing OpenAI models from our country". However, I think LLMs are very good at dealing with problems like QG. Therefore, the experimental results are not convincing enough without comparing with them.
W2. Missing the results of QA4QG-large on SQuAD 1.1.
W3. I don't really understand why generating a full answer is helpful. It would be better to provide detailed intuition or explanation.


**Reproducibility:**

4: Could mostly reproduce the results, but there may be some variation because of sample variance or minor variations in their interpretation of the protocol or method.

**Reviewer Confidence:**

3: Pretty sure, but there's a chance I missed something. Although I have a good feel for this area in general, I did not carefully check the paper's details, e.g., the math, experimental design, or novelty.

---

> ### Author Rebuttal · Authors · 2023-08-29
>
> Thanks for your helpful feedback. We provide the response to your concerns as follows.
>
> **Concern1: LLMs are very good at dealing with problems like QG. The experimental results are not convincing enough without comparing with them.**
>
> Our main idea is to conduct multi-level content planning for QG. In our paper, this idea is realized with medium-sized language models (T5 and MixQG) as base models for full-answer generation and question generation. However, our idea could also be realized with LLM, where both the F-model and Q-model are implemented with LLM. Alternatively, the F-model can be a medium size model where the Q-model is implemented with LLM. Such considerations are nice but too big for a conference paper, given that our analysis has already exceeded the page limit.
>
> Nevertheless, we take your advice and compare MultiFactor with zero-shot GPT-3.5 Turbo. Specifically, we randomly select 1000 samples from the split of Fei [1, 2] test set, in the full-document setting to evaluate LLM’s zero-shot performance on automatic metrics. The results are shown in the following:
>
> |            **Model**           |  **BLEU4**  | **ROUGE-L** |  **METEOR** | **BERTScore** |
> |:------------------------------:|:-----------:|:-----------:|:-----------:|:-------------:|
> |          GPT-3.5-Turbo         |     8.71    |    22.09    |    14.48    |     24.70     |
> |    MultiFactor (MixQG-base)    |    29.12    |     45.20   |    30.01    |     54.49     |
>
> As we can see, MultiFactor outperforms GPT-3.5-Turbo significantly in automatic scores. We further conduct human evaluation on 100 samples by asking humans to pick the better one from the questions generated by MultiFactor and GPT-3.5-Turbo (factual consistency and complexity). If both MultiFactor and GPT-3.5-Turbo outputs contains errors (semantic errors or hop error), we consider those results are Tie. The results are shown in the following:
>
> |           **Preferred**           | **GPT-3.5-Turbo win** |  **Tie**  | **MultiFactor win** |
> |:---------------------------------:|:---------------------:|:---------:|:-------------------:|
> |             100 samples           |           20          |     29    |          51         |
> |     Samples w.o. yes/no answer    |           18          |     23    |          51         |
>
> When the answer is yes/no, it is difficult to generate a comparison question without extra instructions or demos. After removing these (8 samples) and evaluate again, we still found our MultiFactor is better.
>
> [1] Fei, Zichu, et al. CQG: A Simple and Effective Controlled Generation Framework for Multi-Hop Question Generation. ACL, 2022.
>
> [2] Yang, Zhilin, et al. HotpotQA: A Dataset for Diverse, Explainable Multi-hop Question Answering. EMNLP, 2018.
>
> **Concern2: Missing the results of QA4QG-large on SQuAD 1.1.**
>
> In the original paper, QA4QG has not been tested on SQuAD1.1. Furthermore, since the source code is also not accessible, we cannot conduct experiments by ourself. As such, we cannot report results on SQuAD 1.1.
>
>
> **Concern3: What is the intuition of full answer?**
>
> Intuitively, a full answer can be regarded as an answer-aware summary of the context, from which complex questions are more conveniently generated. In other words, generating full answers helps remove noise or irrelevant information from long context, thus benefiting QG. Empirically, removing irrelevant from the context is helpful. This is supported by the fact that baselines on HotpotQA (full documents) performs worse than on HotpotQA (support facts) in Table 9. We have mentioned this in lines 075-085, lines 414-418 and demonstrated with Figure 1. We will improve our writing to make this clearer.

---

### Official Review · Reviewer_uHqT · 2023-08-06

**Typos Grammar Style And Presentation Improvements:** NA
**Soundness:** 3

**Excitement:**

4: Strong: This paper deepens the understanding of some phenomenon or lowers the barriers to an existing research direction.

**Missing References:**

NA

**Paper Topic And Main Contributions:**

This paper proposes to conduct automatic content planning at both the phrase and the sentence levels for generating questions.

At the fine-grained level, answer-aware phrases are selected as the focus for downstream QG. At the coarse-grained level, a full answer generation is trained to connect such (disjointed) phrases and form a complete sentence.

**Questions For The Authors:**

See Reasons To Reject

**Reasons To Accept:**

(1) This paper is easy to understand
(2) Regarding full answer as an answer-aware summary of the context is a very clever approach.
(3) The experiment results seems strong




**Reasons To Reject:**

I'm not sure how to obtain the "full answer" during training.

**Reproducibility:**

4: Could mostly reproduce the results, but there may be some variation because of sample variance or minor variations in their interpretation of the protocol or method.

**Reviewer Confidence:**

4: Quite sure. I tried to check the important points carefully. It's unlikely, though conceivable, that I missed something that should affect my ratings.

---

> ### Author Rebuttal · Authors · 2023-08-29
>
> **Q1: how to obtain the "full answer" during training?**
>
> Given the answer and gold question, we apply QA2D [1] to convert question and answer pairs to obtain pseudo (gold) full answers for training. We have mentioned this in Line 371 - 373.  We will revise the paper to make that clearer.
>
> [1] Demszky, Dorottya, et al. Transforming Question Answering Datasets Into Natural Language Inference Datasets. Stanford University. arXiv, 2018.

---

### Official Review · Reviewer_kFf7 · 2023-08-11

**Soundness:** 3

**Excitement:**

3: Ambivalent: It has merits (e.g., it reports state-of-the-art results, the idea is nice), but there are key weaknesses (e.g., it describes incremental work), and it can significantly benefit from another round of revision. However, I won't object to accepting it if my co-reviewers champion it.

**Paper Topic And Main Contributions:**

This paper proposed a novel QG framework, MultiFactor, based on multi-level content planning for question generation. The MultiFactor includes two components: FA-model, which take the context and short answer as input to select key phrases and generates full answers simultaneously. Then, the second part Q-model which takes the generated full answer as an additional input to generate questions.  Experimental results show that their method outperforms previous baselines on two popular QG datasets.

**Questions For The Authors:**

1. For your HotpotQA (full) setting, you used the distractor setting or fullwiki setting, if fullwiki setting, how did you retrieve the full document context?

**Reasons To Accept:**

1. This paper propose to use Phrase-Enhanced Transformer (PET) to realize the MultiFactor, in which the FA-model and Q-model share the same transformer encoder  enabling better representation learning for both phrase selection and generation tasks.
2. Comparing with previous work selecting one sentence from the raw context to reduce the context length, this MultiFactor propose to generate a long answer sentence based on the whole context and recognized named entity or phrase.
3. The experiments results show that the MultiFactor can outperform than many previous baselines on two popular QG datasets.

**Reasons To Reject:**

1. The MultiFactor rely on the named entities or phrases from the context which are recognized in advance by using SpaCy. So I have two concerns here: 1. the performance of NER will influence the final results a lot. 2. This method, MultiFactor, seems only focus on generating questions whose answer is short phrase or one entity so that it can not generate some complex questions, such non-factoid questions.
2. This paper lack of comparing their MultiFactor with some strong baselines, such as zero-shot LLMs. Maybe the authors don't need to compare with ChatGPT, but I think GPT2 or GPT3 are needed.


**Reproducibility:**

4: Could mostly reproduce the results, but there may be some variation because of sample variance or minor variations in their interpretation of the protocol or method.

**Reviewer Confidence:**

5: Positive that my evaluation is correct. I read the paper very carefully and I am very familiar with related work.

---

> ### Author Rebuttal · Authors · 2023-08-29
>
> We appreciate your constructive suggestions. We provide our response to your questions and concerns as follows.
>
> **Concern1: The performance of NER will influence the final results a lot.**
>
> The performance of named entity recognition or phrase recognition has been improved significantly in recent years. In fact, previous methods such as DP-Graph [1], MulQG [2], IGND [3], JointGen [4], and CQG [5], also used named entity from spacy or other toolkits as the preprocessing stage for their entity graph construction. The current challenge still lies in handling disjointed facts in a long context. Our solution of an answer-aware summary (full answers) could help mitigate this challenge.
>
>
> **Concern2: This method, MultiFactor, seems only focus on generating questions whose answer is short phrase or one entity so that it can not generate some complex questions, such non-factoid questions.**
>
> The focus on the task of factoid question generation allows for deeper analysis. Note that our research is in line with many previous research on QG, such as DP-Graph [1], MulQG [2], IGND [3], JointGen [4], and CQG [5], etc. Notably, CQG only focuses on HotpotQA supporting facts setting, whereas we conduct experiments on two settings of HotpotQA and SQuAD for deeper analysis. However, we do agree that the summary-to-QG is an interesting approach to non-factoid question generation, which we intend to explore in the near future.
>
>
> **Concern3: This paper lack of comparing their MultiFactor with some strong baselines, such as zero-shot LLMs.**
>
> Our main idea is to conduct multi-level content planning for QG. In our paper, this idea is realized with medium-sized language models (T5 and MixQG) as base models for full-answer generation and question generation. However, our idea could also be realized with LLM, where both the F-model and Q-model are implemented with LLM. Alternatively, the F-model can be a medium size model where the Q-model is implemented with LLM. Such considerations are nice but too big for a conference paper, given that our analysis has already exceeded the page limit.
>
> Nevertheless, we take your advice and compare MultiFactor with zero-shot GPT-3.5 Turbo. Specifically, we randomly select 1000 samples from the split of Fei [5, 6] test set, in the full-document setting to evaluate LLM’s zero-shot performance on automatic metrics. The results are shown in the following:
>
> |            **Model**           |  **BLEU4**  | **ROUGE-L** |  **METEOR** | **BERTScore** |
> |:------------------------------:|:-----------:|:-----------:|:-----------:|:-------------:|
> |          GPT-3.5-Turbo         |     8.71    |    22.09    |    14.48    |     24.70     |
> |    MultiFactor (MixQG-base)    |    29.12    |     45.20   |    30.01    |     54.49     |
>
> As we can see, MultiFactor outperforms GPT-3.5-Turbo significantly in automatic scores. We further conduct human evaluation on 100 samples by asking humans to pick the better one from the questions generated by MultiFactor and GPT-3.5-Turbo (factual consistency and complexity). If both MultiFactor and GPT-3.5-Turbo outputs contains errors (semantic errors or hop error), we consider those results are Tie. The results are shown in the following:
>
> |           **Preferred**           | **GPT-3.5-Turbo win** |  **Tie**  | **MultiFactor win** |
> |:---------------------------------:|:---------------------:|:---------:|:-------------------:|
> |             100 samples           |           20          |     29    |          51         |
> |     Samples w.o. yes/no answer    |           18          |     23    |          51         |
>
> When the answer is yes/no, it is difficult to generate a comparison question without extra instructions or demos. After removing these (8 samples) and evaluate again, we still found our MultiFactor is better.
>
>
> **Q1: For your HotpotQA (full) setting, you used the distractor setting or fullwiki setting, if fullwiki setting, how did you retrieve the full document context?**
>
> We follow the MulQG [2], GATENLL+CT [7], and QA4QG [8] experimental settings. It is distractor setting, which provides the ground-truth paragraph directly instead of retrieve from the documents pooling.
>
> [1] Pan, Liangming, et al. Semantic Graphs for Generating Deep Questions. ACL, 2020.
>
> [2] Su, Dan, et al. Multi-Hop Question Generation with Graph Convolutional Network. EMNLP, 2020.
>
> [3] Fei, Zichu, et al. Iterative GNN-Based Decoder for Question Generation. EMNLP, 2021.
>
> [4] Cao, Shuyang, et al. Controllable Open-Ended Question Generation with A New Question Type Ontology. ACL, 2021.
>
> [5] Fei, Zichu, et al. CQG: A Simple and Effective Controlled Generation Framework for Multi-Hop Question Generation. ACL, 2022.
>
> [6] Yang, Zhilin, et al. HotpotQA: A Dataset for Diverse, Explainable Multi-hop Question Answering. EMNLP, 2018.
>
> [7] Sachan, Devendra Singh, et al. Stronger Transformers for Neural Multi-Hop Question Generation. arXiv, 2020.
>
> [8] Su, Dan, et al. QA4QG: Using Question Answering to Constrain Multi-Hop Question Generation. ICASSP, 2022.

---

### Official Review · Reviewer_MLhS · 2023-08-12

**Soundness:** 3

**Excitement:**

3: Ambivalent: It has merits (e.g., it reports state-of-the-art results, the idea is nice), but there are key weaknesses (e.g., it describes incremental work), and it can significantly benefit from another round of revision. However, I won't object to accepting it if my co-reviewers champion it.

**Paper Topic And Main Contributions:**

This paper focuses on question generation task. More specifically, it generate questions from long context in 2-stage model. In first stage, it  learns jointly to select keyphrases and generate a supporting sentence from given the context and answer, followed by the second stage of generating target question by using  generated supporting sentence along with the given context as well as answer. The idea behind generating supporting sentence instead of extracting relevant sentence is that generating sentence can summarize relevant context information well.  In both generation steps, a new phrase-enhanced attention mechanism has been introduced here.

**Questions For The Authors:**

1. How to determine which phrases from context are important to the answer or target question in both cases: while generating supporting sentence as well as target question?
2. From the experimental results, using the generate full answer sentence seems marginal impact on performance across all metrics and datasets. If context is too enough, generating another sentence might be overkilling. It would be better to see the performance of model (Du and Cardie (2017) which use the important sentences extraction instead generating sentence in case of long context. The missing result of this baseline make the impact of answer sentence generation questionable.
3. Instruction-tuned model like FlanT5-x model shows better performance in QA or QG tasks compared to T5 models. They shows more capability over reasoning. It would be  better to see the performance of instruction-tuned model as baseline in question generation.

**Reasons To Accept:**

1. The idea of using the generative property of LLM to summarize the target-relevant information then use it to generate target question is one of the strongest points of this paper.
2. Secondly, using the phrase-probabilities to augment key values in transformer attention layers is a notable way to help decoder make use of factual information.

**Reasons To Reject:**

1. The significance of generating full sentence doesn't much reflect in experimental results as Table 9 as well as Table 3. It is not clear that how the way of preparing ground truth answer sentence effectively serve the main purpose of summarizing the answer-relevant information. This may question one of major contributions in this paper. In addition, the absence of QG model with content selection in extractive way in comparison doesn't make paper's position strong.
2. Few concerns are listed in the questions for the authors section.

**Reproducibility:**

5: Could easily reproduce the results.

**Reviewer Confidence:**

4: Quite sure. I tried to check the important points carefully. It's unlikely, though conceivable, that I missed something that should affect my ratings.

---

> ### Author Rebuttal · Authors · 2023-08-29
>
> We appreciate your constructive feedback. We provide our response to your questions and concerns as follows.
>
> **Q1: How to determine which phrases from context are important to the answer or target question in both cases: while generating supporting sentence as well as target question?**
>
> As for the phrase-level planning, we follow the previous works’ setting, such as  DP-Graph [1], IGND [3], JointGen [4], and CQG [5], to first select a pool of candidate phrases from context using off-the-self NER and phrase selection tools. We then train PET models to simultaneously select phrases and generate full answers or questions. During training, if a phrase appears in the gold question by the crowdsource, we consider it as a positive phrase. Please refer to line 338 - 346 for more information, we will make it clearer in the later version.
>
> [1] Pan, Liangming, et al. Semantic Graphs for Generating Deep Questions. ACL, 2020.
>
> [2] Fei, Zichu, et al. Iterative GNN-Based Decoder for Question Generation. EMNLP, 2021.
>
> [3] Cao, Shuyang, et al. Controllable Open-Ended Question Generation with A New Question Type Ontology. ACL, 2021.
>
> [4] Fei, Zichu, et al. CQG: A Simple and Effective Controlled Generation Framework for Multi-Hop Question Generation. ACL, 2022.
>
>
> **Concern1: The significance of generating full sentence doesn't much reflect in experimental results as Table 9 as well as Table 3. It is not clear that how the way of preparing ground truth answer sentence effectively serve the main purpose of summarizing the answer-relevant information.**
>
> Answer relevant summary helps remove unrelevant information from context and facilitates question generation, particularly from long context. In our model, full answer generation attempts to generate an abstractive summary relevant to the given (short) answer. The role of full answer is supported by the significant improvement of MultiFactor over PET-Q on HotpotQA (full document setting) in Table 9. We admit the marginal improvement of automatic metrics in Table 3 and on SQuAD dataset in Table 9. However, it does not mean generating full answers does not help. It could just mean that the automatic evaluation metrics are limited. From human evaluation in Table 5, we can see that MultiFactor is significantly better than Pet-Q, particularly for bridge questions that require multi-hop information. Additionally, better performance is expected for better full answers, as shown in the gap between MultiFactor and Oracle-FA in Table 5. We will make our discussion clear.
>
> **Concern2: The absence of QG model with content selection in extractive way in comparison doesn't make paper's position strong. It would be better to see the performance of model (Du and Cardie, 2017) which use the important sentences extraction instead generating sentence in case of long context. The missing result of this baseline make the impact of answer sentence generation questionable.**
>
> It would be unfair to compare with Du and Carie [1] since this work relies on LSTM as base models. However, in HotpotQA, the supporting facts are the answer-relevant sentences marked by crowdsourcing. As a result, we could treat systems on HotpotQA [2] (support facts) as those that make use of (gold) extractive summary (like Du and Carie) instead of full answer generation like ours. We can cross-reference the performance in Table 2 to see the effectiveness of full answer generation. Specifically, state-of-the-art systems on HotpotQA (support facts) such as CQG[3], MixQG-base[4], QA4QG[5], and PET-Q do not work better than MultiFactor on Hotpot QA (full documents). For example, QA4QG-large on HotpotQA (support facts) obtains Bleu-1 of 49.55, much lower than Multifactor (MixQG-base) with 54.84 of Blue-1. Note that we can conduct such a comparison because the test set is the same, and only the input context is different in the two settings of HotpotQA. We have briefly discussed this from Line 414 to Line 424, but we will clarify it in the final version.
>
> [1] Du, Xinya, et al. Learning to Ask: Neural Question Generation for Reading Comprehension. ACL, 2017.
>
> [2] Yang, Zhilin, et al. HotpotQA: A Dataset for Diverse, Explainable Multi-hop Question Answering. EMNLP, 2018.
>
> [3] Fei, Zichu, et al. Iterative GNN-Based Decoder for Question Generation. EMNLP, 2021.
>
> [4] Murakhovs'ka, Lidiya, et al. MixQG: Neural Question Generation with Mixed Answer Types. NAACL, 2022.
>
> [5] Su, Dan, et al. QA4QG: Using Question Answering to Constrain Multi-Hop Question Generation. ICASSP, 2022.
>
>
> **Q3: Instruction-tuned model like FlanT5-x model shows better performance in QA or QG tasks compared to T5 models. They shows more capability over reasoning. It would be better to see the performance of instruction-tuned model as baseline in question generation.**
>
> Thank you for your suggestions. We have conducted our experiments with two base LM, T5, and MixQG which is finetuned from T5-x for better QG. The experiments aim to verify the effectiveness of our multi-level content planning. As such, adding another base model like Flan-T5 is nice-to-have but may not bring more insights from the perspective of our study. In addition, previous baselines such as QA4QG also do not rely on instruction finetuning models. Hence, doing so may not be fair.

---

### Meta-Review · Area_Chair_679r · 2023-09-20

**Recommendation:** 3

**Metareview:**

The paper proposes a multi-level content planning approach for generating complex questions, specifically at both the phrase and sentence levels. Experiments, including ablation studies, conducted on standard datasets, i.e., HotpotQA and SQuAD, show the effectiveness of the proposed solution. The paper is well-structured and easy to follow, and its findings may be of interest to the QA community.

---

### Decision · Program_Chairs · 2023-10-07

**Decision:**

Accept-Findings

**Comment:**

The paper proposes a multi-level content planning approach for generating complex questions, specifically at both the phrase and sentence levels. Experiments, including ablation studies, conducted on standard datasets, i.e., HotpotQA and SQuAD, show the effectiveness of the proposed solution. The paper is well-structured and easy to follow, and its findings may be of interest to the QA community.